

# CCL25/CCR9 interaction promotes the malignant behavior of salivary adenoid cystic carcinoma *via* the PI3K/AKT signaling pathway

Songling Chai[1,2,*], Zhihao Wen[1,*], Rongxin Zhang[3], Yuwen Bai[1], Jing Liu[1], Juanjuan Li[1], Wenyao Kongling[1], Weixian Chen[1], Fu Wang[1,4] and Lu Gao[1,4]

[1] School of Stomatology, Dalian Medical University, Dalian, China
[2] The Affiliated Stomatological Hospital of Dalian Medical University, Dalian Medical University, Dalian, China
[3] Department of Dermatology, The Second Hospital of Dalian Medical University, Dalian, China
[4] Academician Laboratory of Immune and Oral Development & Regeneration, Dalian Medical University, Dalian, China
* These authors contributed equally to this work.

Corresponding authors
Fu Wang, fuwang@dmu.edu.cn
Lu Gao, dayigaolu@163.com

## ABSTRACT

**Background:** CC chemokine receptor 9 (CCR9), an organ-specific chemokine receptor, interacts with its exclusive ligand CCL25 to promote tumor proliferation and metastasis. However, the effect of CCR9 on salivary adenoid cystic carcinoma (SACC) malignant behavior remains unknown. This study aimed to investigate the specific molecular mechanism by which CCR9/CCL25 modulates malignant progression in SACC.

**Methods:** Immunohistochemistry staining and RT–qPCR analyses were performed to detect the correlation of CCR9 expression and tumor progression-associated markers in SACC. *In vitro*, SACC cell proliferation and apoptosis were evaluated using Cell Counting Kit-8 and colon formation, and cell migration and invasion were detected by wound healing and transwell assays. Vercirnon was used as an inhibitor of CCR9, and LY294002 was used as an inhibitor of the PI3K/AKT pathway in this study. Western blot and RT–qPCR assays were carried out to measure the downstream factors of the interaction of CCL25 and CCR9. The effect of CCL25 on the development of SACC *in vivo* was examined by a xenograft tumor model in nude mice following CCL25, Vercirnon and LY294002 treatment.

**Results:** CCR9 was highly expressed in SACC compared with adjacent salivary gland tissues, and its level was associated with tumor proliferation and metastases. CCL25 enhanced cell proliferation, migration, and invasion through its interaction with CCR9 and exerted an antiapoptotic effect on SACC cells. Targeting CCR9 *via* Vercirnon significantly reduced the phosphorylation level of AKT induced by CCL25. CCL25/CCR9 could activate its downstream factors through the PI3K/AKT signaling pathway, such as cyclin D1, BCL2 and SLUG, thus promoting SACC cell proliferation, antiapoptosis, invasion and metastasis. The *in vivo* data from the xenograft mouse models further proved that CCL25 administration promoted malignant tumor progression by activating the PI3K/AKT pathway.

**Conclusion:** The interaction of CCL25 and CCR9 promotes tumor growth and metastasis in SACC by activating the PI3K/AKT signaling pathway, offering a promising strategy for SACC treatment.

# INTRODUCTION

Salivary adenoid cystic carcinoma (SACC) is a common salivary gland tumor, accounting for approximately 30% of salivary gland malignant tumors in the oral and maxillofacial regions (*Chau et al., 2012*). The biological characteristic of SACC is susceptibility to infiltrating peripheral nerves and metastasizing to distant tissues, mainly to lung, bone, liver and other organs. In severe cases, it often leads to facioplegia, which greatly affects the patient's quality of life (*Chau et al., 2012*; *Wang et al., 2018*). Clinically, the main treatments for SACC are surgery and radiotherapy. Approximately 70% of patients suffer different degrees of local recurrence and distant metastasis of cancer nests over the whole course of the disease. Therefore, a better understanding of specific molecular targets for the SACC malignant process is urgently needed.

Chemokine receptor 9 (CCR9) is a small transmembrane protein with seven regions belonging to the G protein-coupled receptor (GPCR) family. CCR9 is mainly expressed on the surface of immune cells and intestinal cells and participates in the differentiation of immature T lymphocytes and some inflammatory reactions (*Pathak & Lal, 2020*; *Wu et al., 2021*). Chemokine ligand 25 (CCL25) is the only ligand of CCR9 and is mainly secreted by thymic epithelial cells and intestinal epithelial cells. It plays a key role in mobilizing CCR9-positive cells' directional migration and homing to target tissues (*Chen et al., 2012*; *Svensson et al., 2002*). In recent years, studies have found that CCR9 and CCL25 are highly expressed in some tumors and promote their malignant progression. The CCL25/CCR9 interaction modulates the domain of the G protein, resulting in dissociation and release of the $G_{\beta/\gamma}$ subunit, which can convert the substrate phosphatidylinositol 4,5 diphosphate (PIP2) into phosphatidylinositol 3 phosphate (PIP3) and then phosphorylate the N-terminal PH domain of AKT. Continuous activation of the AKT signaling pathway can promote proliferation and inhibit apoptosis in tumor cells (*Hoxhaj & Manning, 2020*; *Kim et al., 2012*; *Sharma et al., 2010*). On the other hand, studies have shown that in breast cancer, CCL25 increases the expression of matrix metalloproteinases (MMPs), including MMP-2, MMP-3, MMP-9 and MMP-10, which effectively degrade the extracellular matrix (ECM) and promote the invasion of cancer cells (*Johnson-Holiday et al., 2011*). For SACC, many studies have shown that MMP2 and MMP9 were highly expressed both in the tumor and stromal compartments, and their expression is closely related to the malignancy of SACC, indicating that a variety of signal molecules could promote the invasion and metastasis of SACC by regulating MMP2/9 (*Yang et al., 2012*; *Zhou et al., 2014*). Moreover, CCR9 can initiate epithelial-mesenchymal transition (EMT) by activating Wnt/β-catenin pathways to promote osteosarcoma metastasis (*Kong et al., 2021*). EMT is associated with

tumor initiation, invasion, metastasis, and resistance to therapy (*Pastushenko & Blanpain, 2019*). SLUG is a key transcription factor of EMT, which can be mediated by the AKT-GSK3β signaling pathway to promote tumor invasion (*Dai et al., 2019*). In colorectal cancer, AKT phosphorylation can activate the transcription factors SLUG, SNAIL1 and TWIST, leading to increased expression of vimentin and suppression of E-cadherin, which creates suitable conditions for tumor metastasis (*Wei et al., 2020*).

The interaction of CCR25/CCR9 has been proved to be closely related to malignant progression of tumors, but its molecular mechanism in SACC has not been studied. SACC typically grows slowly compared with other carcinomas. However, local recurrence and distant metastasis of SACC is quite common after primary tumor resection. Once metastasis occurs, the median duration of SACC survival is about 3 years. Although complete surgery and radiotherapy have been shown to improve long-term survival, the prognosis of SACC remains poor (*Dillon et al., 2016*; *Li et al., 2017*). In view of the ineffectiveness of cytotoxic chemotherapy in advanced SACC, the targeted therapies are expected to be a potential strategy.

In this study, we explored the functional effect of the CCL25/CCR9 interaction and investigated its downstream molecular signaling pathway on the proliferation, apoptosis, invasion, and metastasis of SACC cells. On this basis, we further clarified the therapeutic effectiveness of targeting CCL25/CCR9 in a SACC xenograft model *in vivo*, which provides a potential therapeutic target for suppressing SACC malignant progression.

# MATERIALS AND METHODS

## Chemicals and antibodies

The main reagents involved in this study were CCL25 (#9046-TK-025/CF; R&D Systems Inc., Minneapolis, MN, USA), CCR9 inhibitor (Vercirnon, #GSK-1605786; MedchenExpress, Monmouth Junction, NJ, USA) and PI3K inhibitor (LY294002; Beyotime Biotechnology, Shanghai, China). For immunohistochemical (IHC) staining and Western blotting (WB), the main primary antibodies are listed in Table S1.
The corresponding horseradish peroxidase (HRP)-conjugated secondary antibody and the 3% hydrogen peroxide ($H_2O_2$) solution used in Western blotting and IHC were purchased from ZSGB Biotech (Beijing, China).

## Patient samples

Thirty paraffin-embedded SACC specimens and ten adjacent normal tissues were obtained from pathologically confirmed SACC patients at the Second Affiliated Hospital of Dalian Medical University (Dalian, China) collected from October 2015 to September 2020. This study was approved by the Ethics Association of Dalian Medical University, and written informed consent was obtained from all study participants. Detailed information on the patients is shown in Table S2.

## Immunohistochemistry (IHC)

The immunohistochemical experiment was performed according to our previous study (*Bai et al., 2021*). All procedures were performed according to the guidelines of the

National Institutes of Health regarding the use of human tissues. First, the 4 μm sections were baked at 63 °C for 2 h, dewaxed in xylene and rehydrated in gradient alcohol. After performing antigen retrieval, sections were treated with 3% hydrogen peroxide ($H_2O_2$) for 10 min and incubated with normal goat serum (ZSGB Biotech, Beijing, China) for 1 h at room temperature. Next, tissue sections were incubated overnight at 4 °C with the following primary antibodies: anti-CCR9 (1:100), anti-E-cadherin (1:200), anti-vimentin (1:200), anti-Ki67 (1:1,000), and anti-SLUG (1:50). Then, antibody binding was detected by horseradish peroxidase streptavidin (#SP-9000; ZSGB Biotech, Beijing, China). Finally, the sections were developed using diaminobenzidine (DAB, #ZLI-9018; ZSGB Biotech, Beijing, China) and counterstained with hematoxylin. The images of staining sections were photographed using a phase microscope (Olympus Corporation, Tokyo, Japan), and the integral optical density (IOD) was evaluated by Image-Pro Plus software (Media Cybernetics, Inc., Rockville, MD, USA).

## Cell culture

SACC cell lines (SACC-83 and SACC-LM) were a kind gift from Prof. Tingjiao Liu, Fudan University, and were cultured with Dulbecco's modified Eagle's medium/nutrient mixture F-12 (DMEM/F12; Gibco, Thermo Fisher Scientific, Inc., Waltham, MA, USA) (*Kong et al., 2019*). Human submandibular gland cell line (HSGs) was purchased from American Type Culture Collection (ATCC, Manassas, VA, USA). HSGs were cultured with Dulbecco's modified Eagle's medium (DMEM, Gibco, Thermo Fisher Scientific, Inc., Waltham, MA, USA) All cells were maintained in medium supplemented with 10% fetal bovine serum (FBS, Gibco, Thermo Fisher Scientific, Inc., Waltham, MA, USA) and 100 U/mL penicillin/100 U/mL streptomycin (Gibco, Thermo Fisher Scientific, Inc., Waltham, MA, USA) at 37 °C in a humidified atmosphere with 5% $CO_2$.

## Quantitative real-time RT–PCR (RT–qPCR)

Total RNA was isolated with TRIzol solution (Invitrogen, Thermo Fisher Scientific, Inc., Waltham, MA, USA). The Quantscript RT kit (Tiangen Biotech, Beijing, China) was used for reverse transcription to convert mRNA into cDNA according to the manufacturer's protocol. The RT–qPCR experiments were performed using Talent qPCR PreMix (SYBR Green; Takara, Otsu, Japan) in a Thermal cycler Dice Real Time System (TP800; Takara, Kyoto, Japan). The difference in mRNA expression was calculated using the $2^{-\Delta\Delta CT}$ method. mRNA expression levels were normalized to GAPDH, and all polymerase reactions were performed in triplicate. The primers involved in RT–qPCR is listed in Table S3.

## Cell proliferation assay

Cell proliferation was measured by Cell Counting Kit-8 (CCK-8; Vazyme Biotech, Nanjing, China). According to the manufacturer's instructions, a 100 μL suspension of cells at a density of $1 \times 10^3$ cells/well was added to a 96-well plate (Corning Incorporated, Corning, NY, USA). When cells attached, 10 μL of CCL25 at the indicated concentrations of 0, 10, 25, 50, 100, 200 and 500 ng/ml was added to each well. After incubating for 24 h or

48 h at 37 °C with 5% $CO_2$, CCK-8 solution was added to each well, and the plate was incubated for 1 h at 37 °C. Finally, the optical density at 490 nm was evaluated by a microplate reader (Flash Spectrum Biotechnology, Shanghai, China).

## Colony formation

$1 \times 10^2$ SACC-83 or SACC-LM cells were seeded in 60 mm dishes and cultured in DMEM/F-12 with 10% FBS and 1% Penicilin-Streptomycin Solution in a humidified incubator at 37°. After adhesion, the cells were treated with 0, 50, 100 and 200 ng/ml CCL25 respectively. Fresh DMEM/F-12 media with 10% FBS were replenished every 5 to 6 days. After 10 or 14 days, the colonies were fixed with 4% PFA at room temperature for 20 min, and then stained with Giemsa stain (Solarbio, Beijing, China) for 15 min. Finally, the dishes were washed with PBS for 3 times and photographed by a phase microscope (Olympus Corporation, Tokyo, Japan).

## Wound healing assay

SACC-LM cells ($1 \times 10^6$) were seeded into six-well plates. When the density of cells grew to 70–80%, 200 μL CCL25 at concentrations of 0, 100, and 200 ng/ml was added to the wells. A 200-μL pipette tip was used to make a straight artificial wound scratch, and the cells were incubated for 24 h at 37 °C. Then, the SACC-LM cells that migrated across this straight scratch were observed by a phase microscope (Olympus Corporation, Tokyo, Japan) and evaluated by ImageJ 1.42 software (National Institutes of Health, Rockville, MD, USA).

## Transwell assays

A 24-well Transwell chamber (8-μm pore size; Corning Incorporated, Corning, NY, USA) coated with or without Matrigel (BD Biosciences Inc., Franklin Lakes, NJ, UAS) was used to detect the migration and invasion of SACC-LM cells. A total of $1 \times 10^4$ cells in 200 μl of FBS-free medium supplemented with 20 μL of CCL25 (0, 100, or 200 ng/ml) were seeded into the upper chamber, and 10% FBS-conditioned medium was placed into the lower chamber. After incubating for 24 h, the cells were fixed with 4% paraformaldehyde (Solarbio Science & Technology, Beijing, China) for 10 min and stained using 0.5% crystal violet solution (Sigma–Aldrich, St. Louis, MO, USA) for 15 min at room temperature. Under a phase microscope, five random visual fields were selected, and the number of penetrated cells was calculated by ImageJ 1.42 software.

## Western blotting (WB)

This experiment was performed according to our previous procedures (*Gao et al., 2020*). Briefly, radioimmunoprecipitation assay lysis buffer (RIPA, Solarbio Science & Technology, Beijing, China) mixed with 1 mM phenylmethylsulfonyl fluoride (PMSF, Beyotime, Shanghai, China) was used to lyse SACC-LM cells on ice, and a bicinchoninic acid protein assay kit (BCA, Beyotime Biotechnology, Shanghai, China) was used to determine the total protein concentration of each sample. Each sample was heated at 96 °C for 5 min to denature the protein, and 20 μg of protein was separated by 10% sodium

dodecyl sulfate polyacrylamide (SDS–PAGE, Beyotime, Shanghai, China) and transferred to a polyvinylidene difluoride membrane (PVDF, Millipore, Merck KGaA, Darmstadt, Germany). Next, the membrane was immersed in 5% defatted milk at room temperature for 1 h and incubated with the corresponding primary antibody (anti-CCR9, 1:500; anti-ERK1/2, 1:1,000; anti-pERK1/2, 1:500; anti-AKT, 1:1,000; anti-p-Ser473-AKT, 1:500; anti-STAT3, 1:1,000; anti-p-Tyr705-STAT3, 1:500; anti-vimentin, 1:1,000; anti-E-cadherin, 1:1,000; anti-Ki67, 1:1,000; anti-MMP2, 1:1,000; anti-MMP9, 1:1,000; anti-BCL2, 1:500; anti-BAX, 1:1,000; anti-cyclin D1, 1:200; and anti-SLUG, 1:500; anti-SNAIL, 1:1,000; anti-TWIST, 1:1,000) at 4 °C overnight. On the following day, the PVDF membrane was rinsed with Tris buffered saline with Tween 20 (TBST) 3 times for 15 min each and then incubated with the secondary HRP-conjugated antibody (1:4,000; ZSGB Biotech, Beijing, China) for 1 h at room temperature. Finally, electrogenerated chemiluminescence solution (ECL, Pierce ECL Plus Western Blotting, Thermo Fisher Scientific, Inc., Waltham, MA, USA) was configured to develop the PVDF membrane. The protein expression was detected using a ProteinSimple FluorChem system (Bio–Rad Laboratories, Hercules, CA, USA) and evaluated by ImageJ 1.42 software.

## Live and dead cell double staining

SACC-LM cells ($1 \times 10^4$) were inoculated in 24-well plates. After attachment, cells were administered with the indicated treatment (PBS, 200 ng/ml CCL25, 10 nM LY294002 and 10 nM Vercirnon) for 24 h. Next, the cells were washed with PBS for 3 times and incubated with Working staining solution (Abbkine, Wuhan, China) at a dose of 400 μL per well at 37 °C for 20 min. After washing with PBS for 3 times, the nuclei were stained with mounting medium containing DAPI for 5 min at room temperature, and cells were observed by a fluorescence microscope (Olympus Corporation, Tokyo, Japan).

## Animals and xenograft formation

Twenty female BALB/c nude mice (18–20 g; 5–6 weeks old) were purchased from the SPF Animal Experiment Center of Dalian Medical University, which all lived in laminar flow cabinets with constant temperature (25–28 °C) and humidity in accordance with the breeding standards. The animal experiment was approved by the Animal Experimental Ethics Committee of Dalian Medical University (AEE20017). First, 100 μl of PBS containing $2 \times 10^6$ SACC-LM cells was subcutaneously inoculated into the mice's right armpits using a sterile No. 22 syringe under anesthesia. After 7 days of injection, the mice were randomly divided into four groups: PBS group (without administration), CCL25 group (administration of 35 ng/kg CCL25), CCL25/Vercirnon group (Vercirnon, CCR9 inhibitor, 50 mg/kg, was injected around the tumor at 7 days and 14 days after CCL25 administration), and CCL25/LY294002 group (LY294002, PI3K inhibitor, 1.5 mg/kg was injected around the tumor at 7 days and 14 days after CCL25 administration). The mouse weight and tumor volume were measured every three days. At 28 days after injection of the tumor cells, the mice were euthanized, and the xenografts were harvested and collected for subsequent experiments, including IHC, TUNEL, WB and RT–qPCR.

## Apoptosis assay

The Colorimetric TUNEL Apoptosis Assay Kit (Vazyme Biotech, Nanjing, China) was used to detect the apoptosis ability of mouse xenograft tumors. According to the manufacturer's instructions and our previous procedure (*Zhang et al., 2020*), the tissues were fixed in 4% paraformaldehyde and embedded in paraffin. Next, the tissues were cut into 4 μm sections and then subjected to conventional deparaffinization and hydration. After drying, the section was incubated with 20 μg/mL proteinase K solution in PBS for 30 min at 37 °C. Fifty microliters of enzyme working solution of terminal deoxynucleotidyl transferase (TdT) was dropped onto the slices to terminate the reaction for 1 h at 37 °C in the dark. Then, the sections were washed with PBS for 5 min and mounted with Anti-Fade mounting medium with DAPI (Beyotime Biotechnology, Shanghai, China). Finally, a fluorescence microscope (Olympus Corporation, Tokyo, Japan) was used to image and detect the staining of sections.

## Statistical analysis

For statistical analysis, using GraphPad Prism 8 (GraphPad Software, La Jolla, CA, USA), one-way analysis of variance followed by Tukey's post-hoc test was analyzed for multiple comparison tests, and two-tailed Pearson's statistics were used for correlation analysis of CCR9 and the markers of tumor proliferation and migration. All experiments were repeated at least 3 times, and the data are shown as the mean ± SD. $P < 0.05$ was considered statistically significant.

# RESULTS

## CCR9 was highly expressed in SACC and correlated with tumor proliferation and invasion

First, we detected the expression and distribution of CCR9 in human SACC by IHC. The results showed that compared with that in adjacent normal tissues, CCR9 was highly expressed in the three types of SACC and was mainly located in the cell membrane and cytoplasm. Interestingly, we found a specific correlation between the level of CCR9 and the different types of SACC. The level of CCR9 was the highest in solid SACC with poor differentiation, while the expression of CCR9 was relatively low in cribriform and tubular SACC with good differentiation (Figs. 1A and B). Next, RT–qPCR was used to detect the mRNA expression of CCR9 in HSGs, SACC-83 and SACC-LM cell lines. As shown in Fig. 1C, the mRNA expression of CCR9 in SACC-83 and SACC-LM cells was significantly higher than that in HSG cells, which was consistent with the IHC results.

To further determine whether the expression level of CCR9 is related to the malignant progression of SACC, IHC was used to detect the expression of proliferation-related antigen Ki67 and migration-invasion related antigen vimentin and E-cadherin in the three types of SACC. The results showed that the expression levels of Ki67 and vimentin in the solid SACC with CCR9 high expression were markedly higher than those in the cribriform and tubular SACC, while the trend of E-cadherin expression in the three types of SACC was the opposite (Fig. 1D). Pearson correlation analysis further proved that CCR9 was positively correlated with the expression of Ki67 and vimentin but negatively correlated

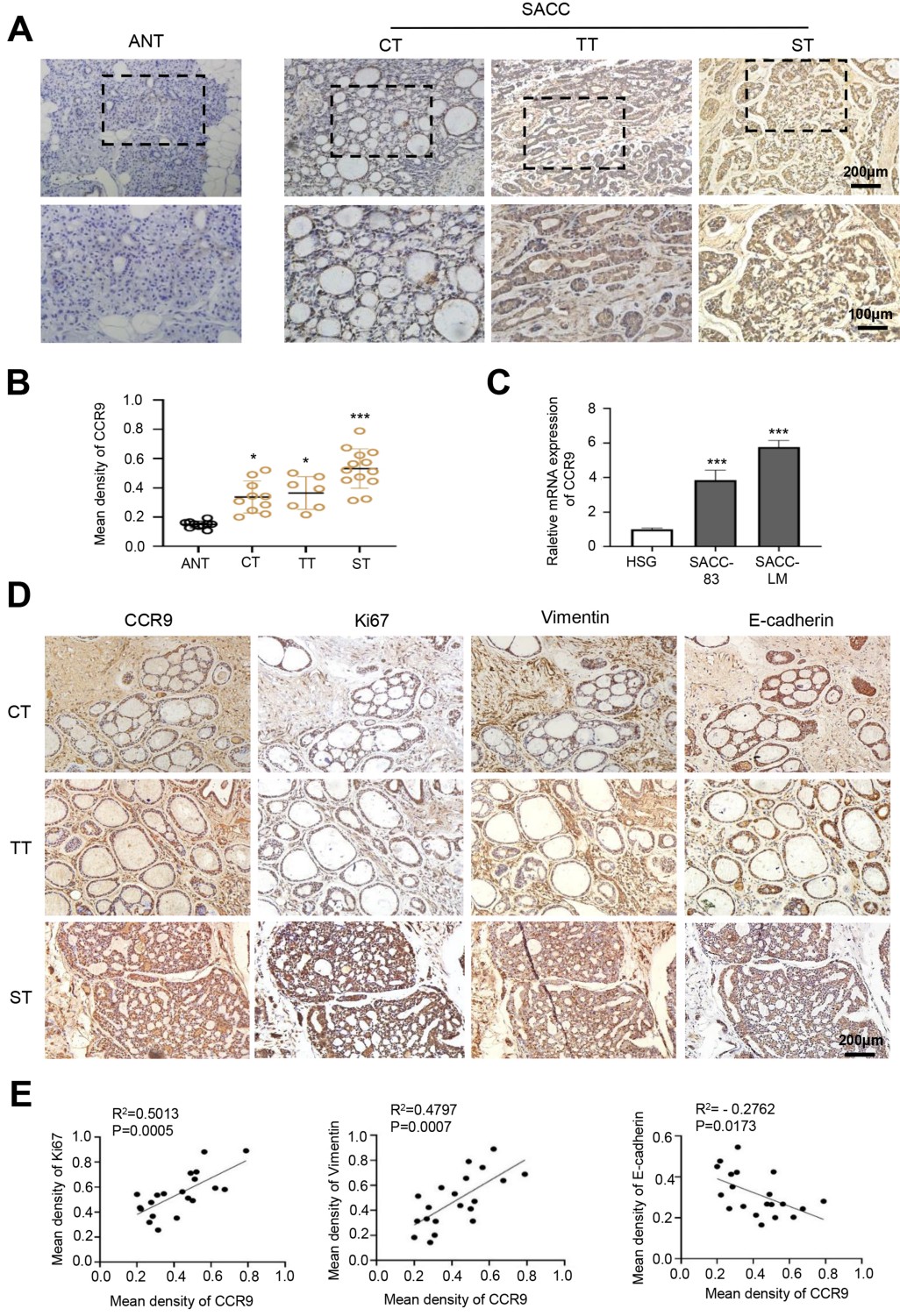

**Figure 1 Expression of CCR9 in SACC and its correlation with tumor proliferation and metastasis.** (A) Representative images showing the expression of CCR9 in different types of SACC (cribriform type, CT, $n = 10$; tubular type, TT, $n = 7$; solid type, ST, $n = 13$) by IHC. Adjacent normal tissue (ANT, $n = 10$) was used as the control group. The boxed areas in the upper panels are shown at higher magnification in the lower panels. (B) The relative fold change in CCR9 expression (A) in SACC was

**Figure 1 (continued)**
quantitatively analyzed (unpaired two-tailed t-test, $^*P < 0.05$, $^{***}P < 0.001$). (C) The mRNA expression of CCR9 was determined by RT–qPCR in the SACC-83, SACC-LM and HSG cell lines ($n = 3$, unpaired two-tailed t-test, $^{***}P < 0.001$). (D) Representative images of the expression of tumor proliferation and invasion markers (Ki67, vimentin, and E-cadherin) by IHC in the three types of SACC. (E) Pearson correlation analysis was used to analyze the expression of Ki67, vimentin, E-cadherin and CCR9 in SACC ($n = 20$, two-tailed Pearson's correlation). Data are presented as the mean ± SD.

with the expression of E-cadherin (Fig. 1E). These results suggested that SACC with high expression of CCR9 has a stronger proliferation and migration ability.

## The CCL25/CCR9 interaction enhanced the proliferation and anti-apoptosis of SACC cells

To test whether CCL25 plays an important role in the proliferation of SACC, we first used a CCK-8 assay to detect the effect of CCL25 on SACC cells. We found that with increasing CCL25 concentration, the proliferation ability of SACC cell lines (SACC-83 and SACC-LM) was increased, especially SACC-LM cells. With the concentration of CCL25 gradient screening, the proliferation ability of SACC cells reached its peak at 200 ng/ml, including 24 h and 48 h (Figs. 2A, 2B). Next, colony formation was used to determine the effects of different concentrations of CCL25 on tumor cells proliferation. The result showed that 100 ng/ml and 200 ng/ml CCL25 significantly enhanced the proliferation of SACC-LM cells, and the proliferation of SACC-LM cells in 200 ng/ml CCL25 group was almost 6 times higher than that in the control group. Therefore, in the following experiment, 200 ng/ml CCL25 was used to treat SACC-LM cells. Moreover, we used a CCR9 inhibitor (Vercirnon, VCN) to further clarify that CCL25 functions through its interaction with CCR9. RT–qPCR and WB was performed to detect the mRNA expression and protein level of proliferation-related factors (cyclin D1, Ki67, c-Myc) and apoptosis-related factors (BCL2, BAX, and caspase 3) in SACC-LM cells. The results showed that compared with the control group, the expression of cyclin D1, Ki67, c-Myc and BCL2 in SACC-LM cells treated with CCL25 was significantly increased, while the expression of BAX and caspase 3 was decreased. When the CCR9 inhibitor was added to the CCL25 treatment group, the expression of the above factors, especially cyclin D1 and BCL2, was reversed (Figs. 2E–2H). This result indicated that the CCL25/CCR9 interaction could promote the proliferation and anti-apoptosis ability of SACC-LM cells, which might be related to the factors cyclin D1 and BCL2.

## The CCL25/CCR9 interaction enhanced SACC cell migration and invasion

To explore the potential effect of CCL25 on the migration and invasion of SACC cells, wound healing and Transwell assays were used to observe the effects of CCL25 treatment on SACC-LM cells. The wound healing results showed that after SACC-LM cells were treated with 100 ng/ml or 200 ng/ml CCL25 in serum-free medium for 24 h, the width of the scratch was significantly decreased compared with that of the control group, especially

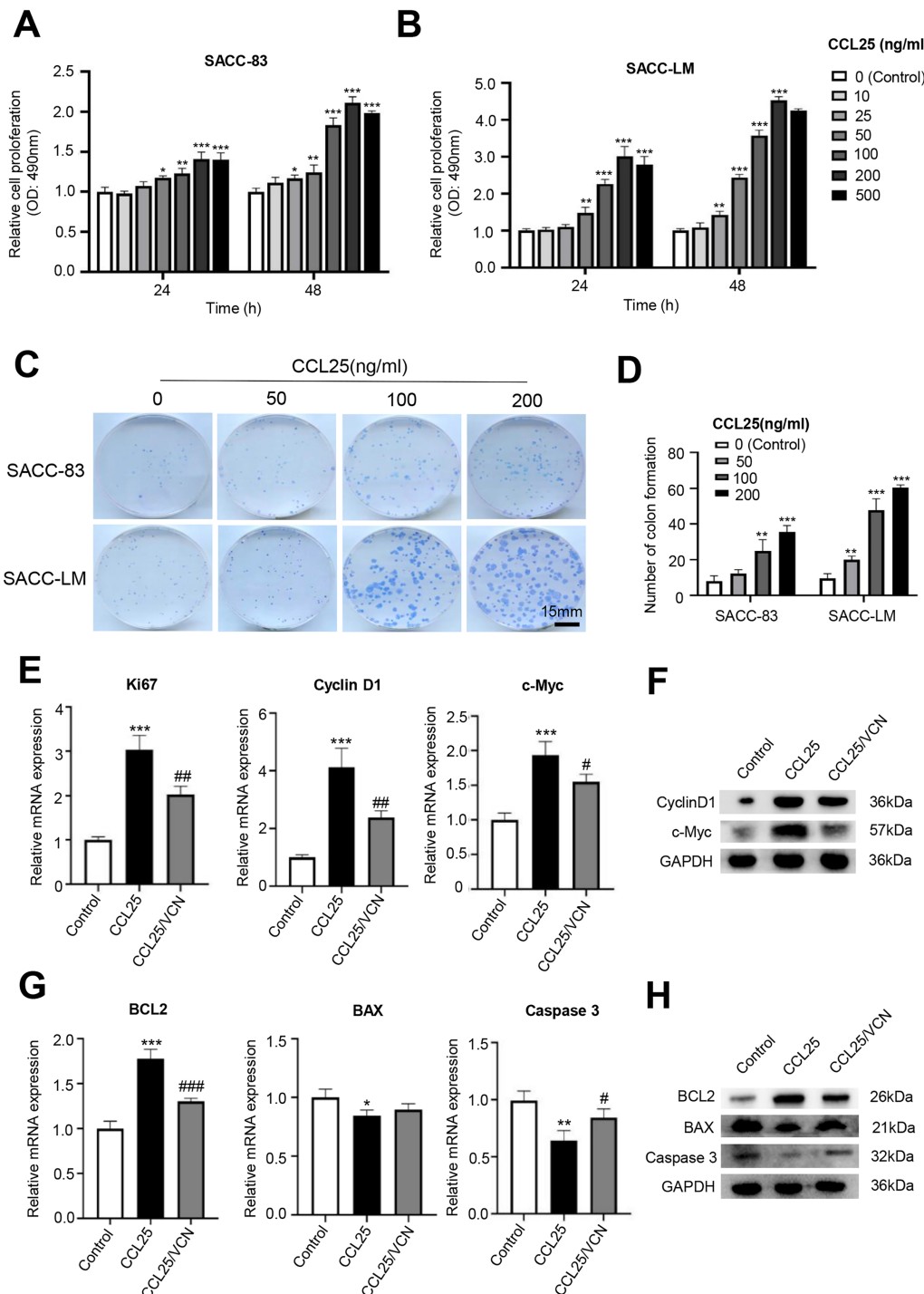

**Figure 2 CCL25/CCR9 promotes the proliferation of SACC cells.** (A, B) The proliferation of SACC-83 (A) and SACC-LM (B) cells treated with different concentrations of CCL25 was tested by a CCK-8 kit at 24 h and 48 h ($n = 3$, unpaired two-tailed t-test; $^{*}P < 0.05$, $^{**}P < 0.01$, $^{***}P < 0.001$). (C) Colony-forming assay was performed to detect the proliferation of SACC-83 and SACC-LM cells treated with CCL25 ($n = 3$, unpaired two-tailed t-test; $^{**}P < 0.01$, $^{***}P < 0.001$). (D) Quantitative analysis of the colon level of (C). (E, F) SACC-LM cells were treated with 200 ng/ml CCL25 and 10 nM CCR9 inhibitor (Vercirnon, VCN) for 48 h. The mRNA and protein expression of proliferation-related factors (Ki67, cyclin D1, and

**Figure 2 (continued)**
c-Myc) was detected by RT–qPCR (E) and WB (F) ($n = 3$, one-way ANOVA followed by Tukey's multiple comparisons; * vs Control, ***$P < 0.001$; # vs CCL25, #$P < 0.05$, ##$P < 0.01$.). (G, H) The mRNA and protein expression of apoptosis-related factors (BCL2, BAX, and caspase 3) was detected by RT–qPCR (G) and WB (H) ($n = 3$, one-way ANOVA followed by Tukey's multiple comparisons; * vs Control, *$P < 0.05$, **$P < 0.01$, ***$P < 0.001$; # vs CCL25, #$P < 0.05$, ###$P < 0.001$). Data are presented as the mean ± SD.                                                      

the 200 ng/ml CCL25 group (Figs. 3A and 3B). The Transwell assay also showed the same trend as the above wound healing assay (Figs. 3C and 3D). These results indicated that CCL25 promoted the migration and invasion of SACC-LM cells in a concentration-dependent manner. In addition, we added a CCR9 inhibitor (Vercirnon) to the culture system. RT–qPCR and WB was used to detect the expression of EMT-related proteins (vimentin and E-cadherin) and ECM degradation-related proteins (MMP2 and MMP9). Compared with the control group, the expression of vimentin, MMP2 and MMP9 in SACC-LM cells treated with 200 ng/ml CCL25 increased markedly, while the expression of E-cadherin decreased. In contrast, when Vercirnon was added to the cells of the CCL25 group, the expression of vimentin, MMP2 and MMP9 was reduced, and the expression of E-cadherin was increased again. The results of WB were consistent with RT-qPCR (Figs. 3E and 3F). EMT plays a key role in the process of invasion and metastasis of epithelial tumor cells (Zhang et al., 2020). These results suggested that CCL25 could cause SACC-LM cells to lose polarity and connections with each other and further enhance the degradation of ECM, thus promoting the migration and invasion of SACC cells.

## CCL25/CCR9 promotes the malignant process of SACC cells by activating the PI3K/AKT signaling pathway

To further investigate whether the interaction between CCL25 and CCR9 was involved in activating signaling pathways related to tumor progression, the phosphorylation levels of the MAPK, PI3K/AKT and JAK/STAT signaling pathways were detected using WB. After treatment with CCL25, the results showed that p-AKT(Ser473) expression was significantly upregulated. When CCR9 was blocked from binding to CCL25 in SACC cells, the level of p-AKT(Ser473) was significantly downregulated, while the phosphorylation level of other signaling molecules did not respond (Figs. 4A–4D). The results suggested that CCL25/CCR9 mainly induced the activation of the PI3K/AKT signaling pathway in SACC cells.

To prove that the ability of CCL25 to promote the proliferation and anti-apoptosis of SACC cells depended on the PI3K/AKT signaling pathway, a PI3K/AKT inhibitor (LY294002) was added to the culture system, and cell proliferation was detected by CCK-8 assay. The results showed that LY294002 could alleviate the proliferation ability of SACC-LM cells promoted by CCL25, and its concentration almost reached saturation at 10 nM (Figs. 5A and 5B). Next, using living and dead cell experiment, we further verified whether CCL25/CCR9 promoted the growth of tumor cells through activating AKT signaling pathway. The results showed that LY294002 could significantly inhibit the growth of SACC-LM cells induced by CCL25, but the number of dead tumor cells did not

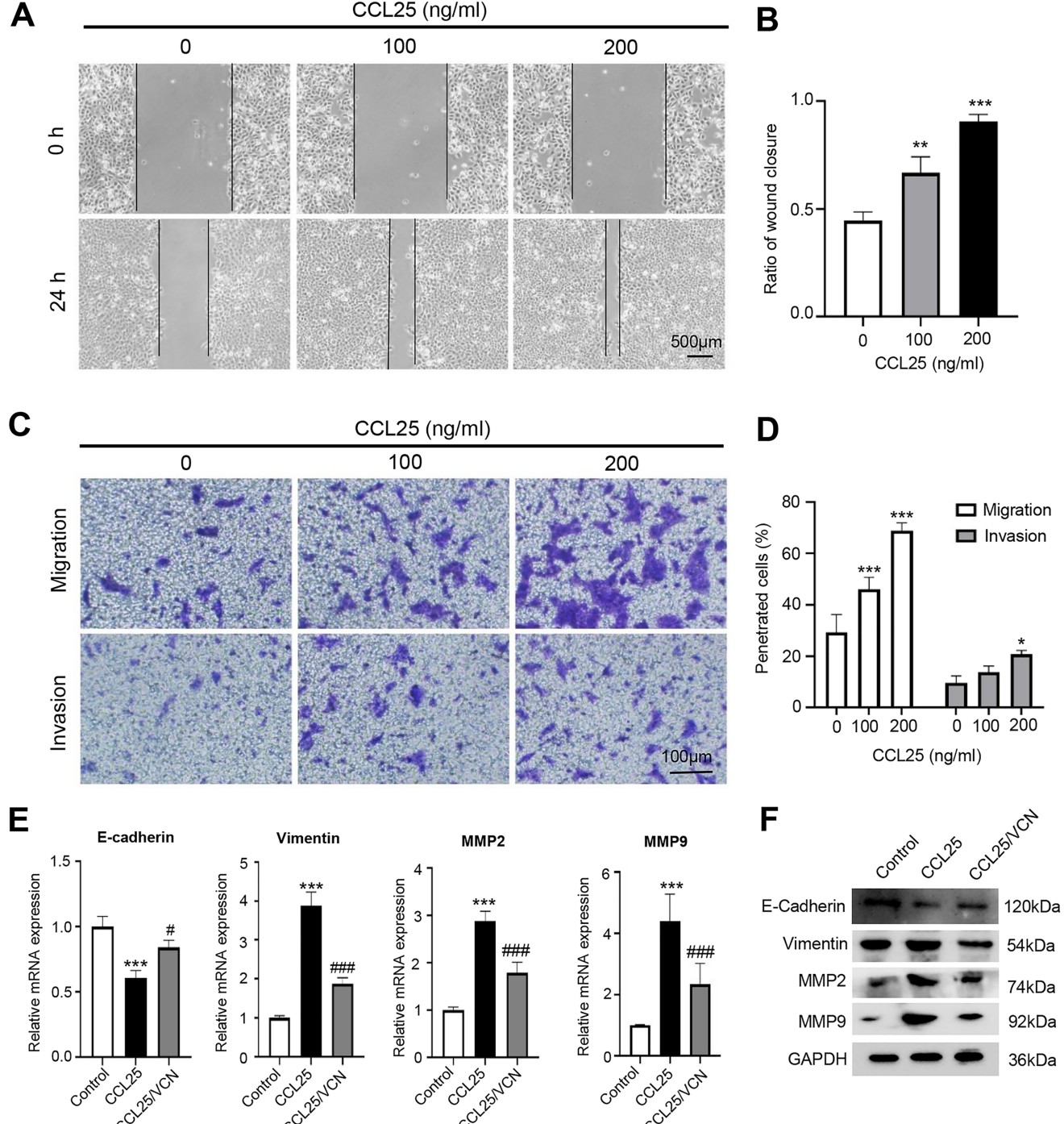

**Figure 3** **CCL25/CCR9 promotes SACC-LM cell migration and invasion.** (A) After SACC-LM cells were treated with different concentrations of CCL25 (0, 100, 200 ng/ml for 24 h, wound healing assays were used to test the horizontal migration ability of SACC-LM cells. (B) The wound closure area of cells after CCL25 administration was quantitatively analyzed ($n = 3$, unpaired two-tailed t-test; $**P < 0.01$, $***P < 0.001$). (C) The migration and invasion abilities were detected by Transwell assays after SACC-LM cells were treated with different concentrations of CCL25 (0, 100, 200 ng/ml) for 24 h. (D) Quantitative analysis of the number of migrated and invasive cells in (C) ($n = 3$, unpaired two-tailed t-test; $*P < 0.05$, $***P < 0.001$). (E, F) The mRNA and protein expression of vimentin, E-cadherin, MMP2 and MMP9 was measured by RT–qPCR (E) and WB (F). GAPDH was used as a standard control ($n = 3$, one-way ANOVA followed by Tukey's multiple comparisons; $*$ $vs$ Control, $***P < 0.001$; $\#$ $vs$ CCL25, $\#P < 0.05$, $\#\#\#P < 0.001$). Data are presented as the mean ± SD.

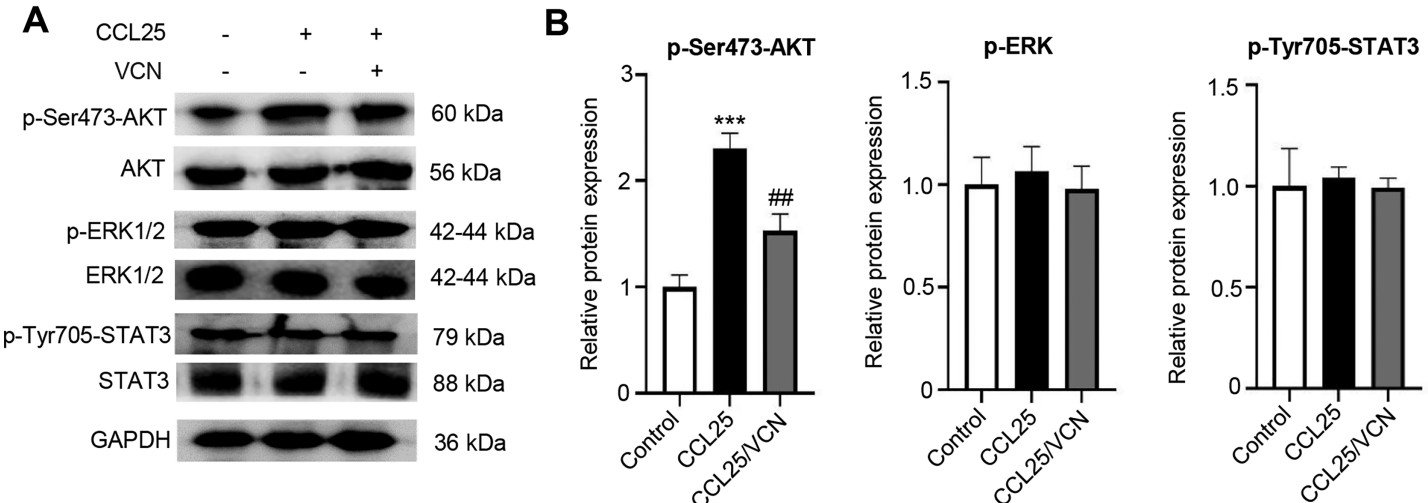

**Figure 4 CCL25/CCR9 activates the PI3K/AKT signaling pathway in SACC-LM cells.** (A) WB was used to detect the activation of signaling pathways, including p-ERK1/2 (ERK1: T202/Y204; ERK2: T185/Y187), p-AKT (Ser473) p-STAT3 (Tyr705) after SACC-LM cells were treated with/without 10 nM Vercirnon for 48 h followed by 200 ng/ml CCL25 pretreatment. (B) Quantitative analysis of the protein level of p-AKT, p-ERK1/2 and p-STAT3 ($n = 3$, one-way ANOVA followed by Tukey's multiple comparisons; ***$P < 0.001$; # vs CCL25, ##$P < 0.01$). Data are presented as the mean ± SD.

increase significantly when CCR9 inhibitor (VCN) was added into the tumor cells with the present of LY294002. In addition, RT–qPCR and WB results showed that compared with the control group, the levels of proliferation-related factors (Ki67, cyclin D1, and c-Myc) and anti-apoptosis-related factors BCL2 in SACC-LM cells treated with CCL25 were significantly increased, and the levels of apoptosis-related factors BAX and caspase 3 were obviously decreased. When the PI3K/AKT inhibitor was added to the CCL25 culture system, the changes in the above factors were restored. Importantly, when CCR9 inhibitor was added in the CCL25/LY294002 group, the level of the above proliferation and apoptosis related factors was not up- or down-regulated significantly, including the expression of mRNA and protein (Figs. 5C–5H). The results indicated that PI3K/AKT was the main signal pathway activated by CCL25/CCR9 in SACC cells, which could enhance the proliferation and anti-apoptosis abilities of tumor cells.

Moreover, we explored whether a PI3K/AKT inhibitor affected the migration and invasion abilities of SACC-LM cells treated with CCL25. The RT–qPCR and WB results showed that the expression of vimentin, MMP2 and MMP9 in the CCL25/LY294002 group was lower than that in the CCL25 group. In contrast, the expression of E-cadherin in the CCL25/LY294002 group was significantly higher than that in the CCL25 group. However, when CCR9 inhibitor (VCN) was added to the CCL25 group with LY294002, vimentin and MMPs did not decrease continuously, and E-cadherin also did not increase (Figs. 6A–6E). The SNAIL family (SNAIL1, SLUG, and TWIST) is the main EMT-related nuclear transcription factor (*Pastushenko & Blanpain, 2019*). RT–qPCR and WB results proved that CCL25 could increase the expression of SNAIL1 and SLUG in SACC-LM cells. When the PI3K/AKT signaling pathway was inhibited, the level of SLUG and SNAIL1 was reversed, especially SLUG. While CCR9 inhibitor could not significantly change the level

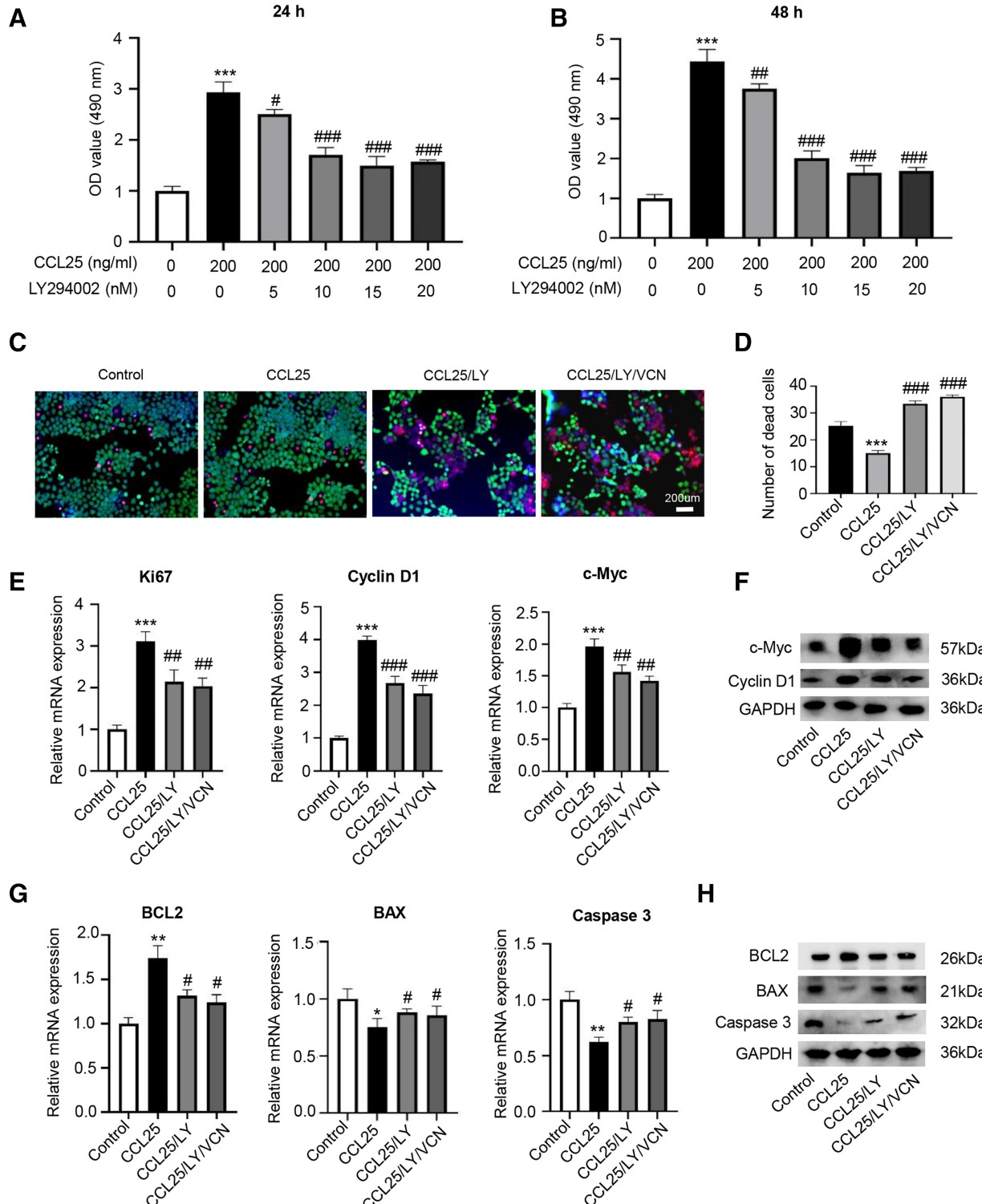

**Figure 5 CCL25/CCR9 interaction enhances the proliferation and anti-apoptosis of SACC-LM cells through the PI3K/AKT pathway.**
(A, B) After treatment with 200 ng/ml CCL25, different concentrations of PI3K/AKT inhibitor (LY294002, LY) were added to $5 \times 10^3$ SACC-LM cells for 24 or 48 h. The proliferation of SACC-LM cells was detected by CCK-8 assay ($n = 3$, one-way ANOVA followed by Tukey's multiple comparisons; $*$ vs Control, $***P < 0.001$; # vs CCL25, #$P < 0.05$, ##$P < 0.01$, ###$P < 0.001$). (C) Cells viability was detected by Live and Dead Cell Double Staining experiment. Red light represented Dead cells and green light represented living cells. (D) Quantitative analysis of the

**Figure 5 (continued)**
fluorescence number of dead cells per unit area. (*n* = 3, one-way ANOVA followed by Tukey's multiple comparisons; * *vs* Control, ****P* < 0.001; # *vs* CCL25, ###*P* < 0.001). (E, F) A total of 1 × 10⁵ SACC-LM cells were administered with 200 ng/ml CCL25, 10 nM LY294002 and 10 nM Vercirnon for 48 h. Proliferation-related factors (Ki67, cyclin D1, and c-Myc) were detected by RT–qPCR (E) and WB (F) (n = 3, one-way ANOVA followed by Tukey's multiple comparisons; * *vs* Control, ****P* < 0.001; # *vs* CCL25, ##*P* < 0.01, ###*P* < 0.001). (G, H) The mRNA and protein expression of apoptosis-related factors (BCL2, BAX, and caspase 3) was detected by RT–qPCR (G) and WB (H) (*n* = 3, one-way ANOVA followed by Tukey's multiple comparisons; * *vs* Control, **P* < 0.05, ***P* < 0.01; # *vs* CCL25, #*P* < 0.05). Data are presented as the mean ± SD.

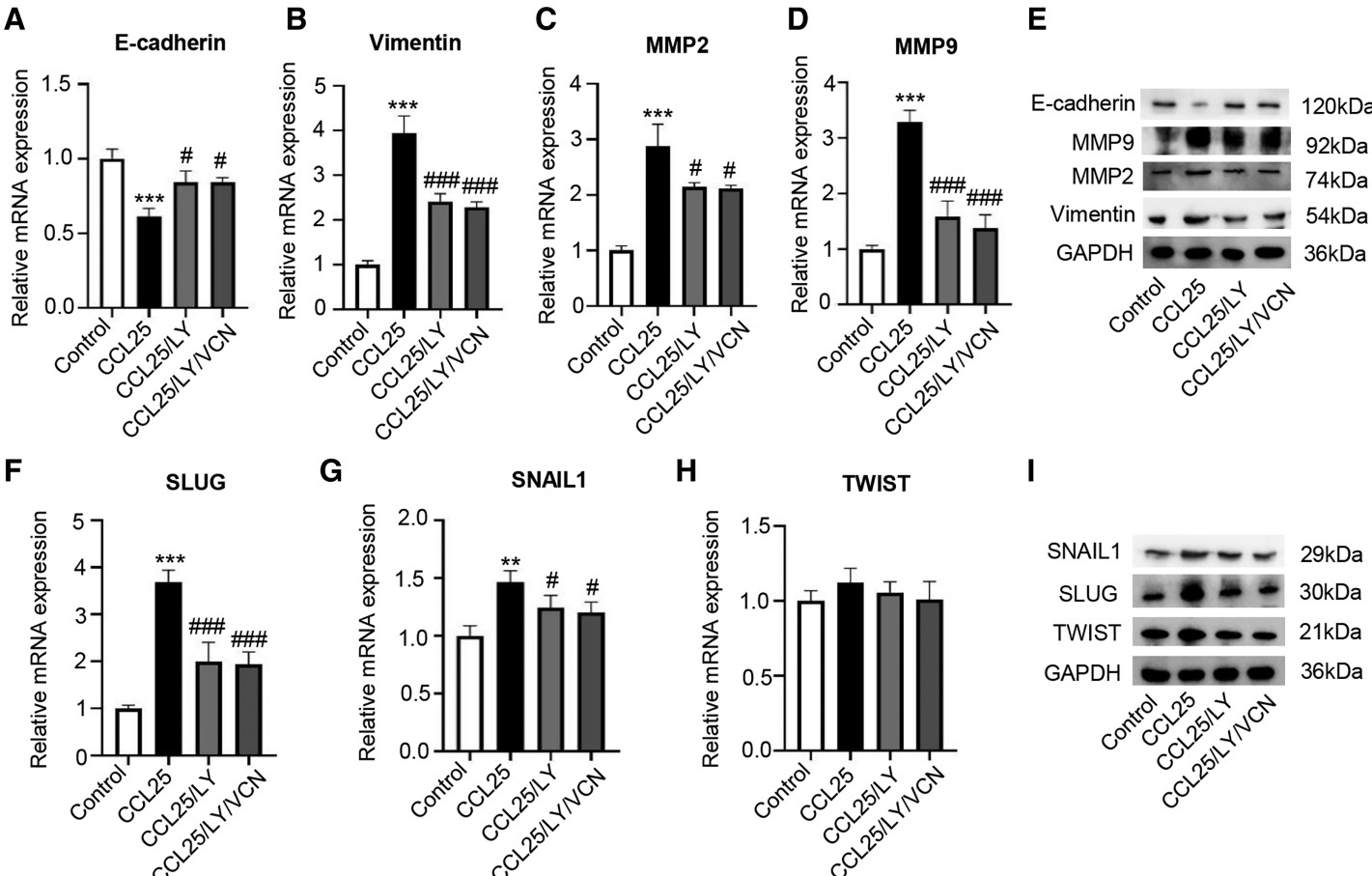

**Figure 6  CCL25/CCR9 interaction enhances the migration and invasion of SACC-LM cells through the PI3K/AKT pathway.** (A–E) A total of 1 × 10⁵ SACC-LM cells were administered 200 ng/ml CCL25, 10 nM PI3K/AKT inhibitor (LY294002) and 10 nM Vercirnon (VCN) for 48 h. The mRNA and protein expression of invasion- and metastasis-related proteins (vimentin, E-cadherin, MMP2 and MMP9) was detected by RT–qPCR (A–D) and WB (E) (n = 3, one-way ANOVA followed by Tukey's multiple comparisons; * *vs* Control, ****P* < 0.001; # *vs* CCL25, #*P* < 0.05, ###*P* < 0.001). (F–I) The mRNA and protein expression of EMT-related transcription factors (SLUG, SNAIL1, and TWIST) was detected by RT–qPCR (F–H) and WB (I) (n = 3, one-way ANOVA followed by Tukey's multiple comparisons; * *vs* Control, ***P* < 0.01, ****P* < 0.001; # *vs* CCL25, ###*P* < 0.001). Data are presented as the mean ± SD.

of the above transcription factors in the present of LY294002 in the CCL25 group (Figs. 6F–6I). These results further indicated that the activation of the PI3K/AKT signaling pathway mediated by CCL25/CCR9 could upregulate the expression of the EMT-related transcription factors and then promote the migration and invasion of SACC cells.

## CCL25 promotes growth and metastasis in a xenograft mouse model *via* the PI3K/AKT signaling pathway

To confirm the above experimental results, we used adult female BALB/c nude mice to construct a xenograft mouse model *in vivo*. The mice were divided into four groups: PBS, CCL25, CCL25/CCR9 inhibitor and CCL25/PI3K/AKT inhibitor groups (Fig. 7A). From the ninth day after injection, the growth rate of tumors in the CCL25 group was significantly faster than that in the PBS group, while the growth rate in the CCL25/Vercirnon and CCL25/LY294002 groups was gradually slower than that in the CCL25 group, and the differences were statistically significant (Figs. 7B and 7C). On the twenty-ninth day, the xenograft tumors in nude mice were harvested. HE staining showed that the density of tumor parenchymal cells in the CCL25 group was higher than that in the inhibitor groups, and tumor stroma in the CCL25 group was rarely seen. At the same time, TUNEL and Ki67 staining showed that compared with the PBS group, most tumor cells were in a state of proliferation and less in apoptosis in the CCL25 group. In the CCL25/Vercirnon and CCL25/LY294002 groups, the number of apoptotic cells was significantly higher than that in the CCL25 group, and the number of proliferating cells showed the opposite trend (Figs. 7D and 7E). WB and RT–qPCR analysis in the four groups further proved that the expression of CyclinD1 and BCL2 in the CCL25 group was significantly higher than that in the CCL25/Vercirnon and CCL25/LY294002 groups (Figs. 7F and 7G). These results revealed that CCL25 promoted the proliferation and anti-apoptosis of SACC cells through the CCR9/PI3K/AKT signaling pathway *in vivo*.

In addition, we used IHC, WB and RT–qPCR assays to detect the expression of tumor invasion- and metastasis-related proteins (E-cadherin, vimentin, and SLUG) in the four groups. The results showed that the expression of vimentin and SLUG in the CCL25 group was distinctly higher than that in the inhibitor groups (Figs. 8A–8E). We found that the effect of inhibiting PI3K/AKT was basically the same as that of directly inhibiting CCL25/CCR9, implying that CCL25/CCR9 can regulate the expression of the EMT-related transcription factor SLUG through the PI3K/AKT signaling pathway, which results were consistent with tumor cells experiments. Next, we measured the expression of MMP2 and MMP9 in tumor tissues. The protein and mRNA levels of MMP2 and MMP9 in the CCL25 group were significantly higher than those in the CCL25/Vercirnon and CCL25/LY294002 groups, especially MMP9, and the results were consistent with SLUG's trend (Figs. 8F–8H). The results suggested that the activation of SLUG by CCL25/CCR9 might further modulate the expression of ECM degradation-related proteins in SACC. In summary, the above results proved that in SACC, the interaction between CCL25 and CCR9 could activate its downstream factors through the PI3K/AKT signaling pathway, such as cyclin D1, BCL2 and SLUG, thereby promoting SACC proliferation, anti-apoptosis, invasion and metastasis (Fig. 9).

## DISCUSSION

Signal transduction mediated by G protein-coupled receptors is highly related to cell proliferation, apoptosis, migration and invasion, cytoskeleton rearrangement and drug dependence (*Zielińska & Katanaev, 2019*). CCR9, a GTPase superfamily member, binds to

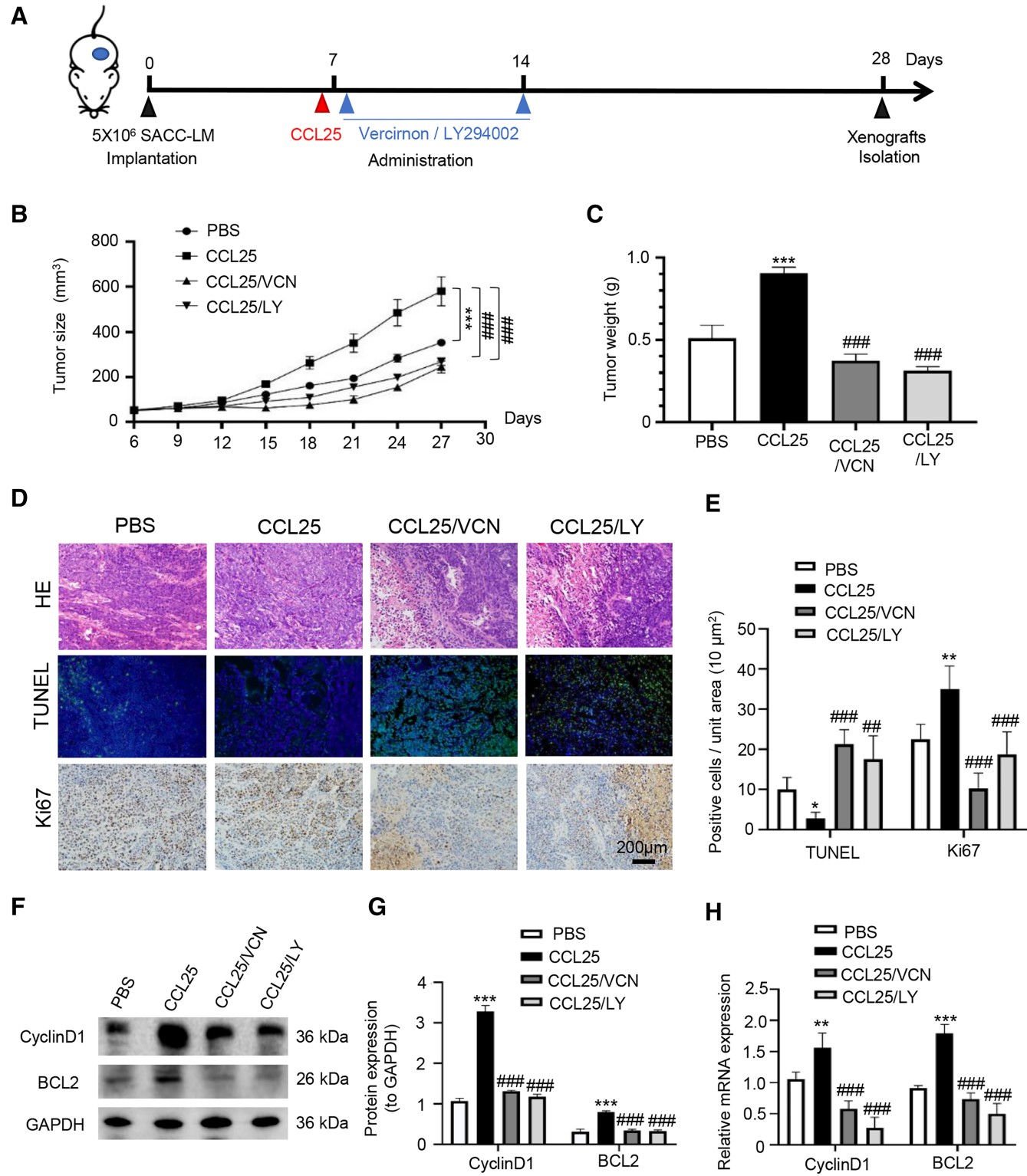

**Figure 7 CCL25/CCR9 promotes tumor growth in a SACC xenograft mouse model by activating the PI3K/AKT signaling pathway.**
(A) Schematic representation of the animal model establishment. For xenograft models, $1 \times 10^7$ SACC-LM cells in 100 µl of PBS were sub-cutaneously injected into the right flanks of mice. After 7 days of tumor cell injection, the mice were randomly divided into four groups: PBS group, CCL25 group, CCL25/VCN group and CCL25/LY group. According to the time axis, mice were administered CCL25 on Day 7 and then given CCR9 inhibitor (Vercirnon) or PI3K/AKT inhibitor (LY294002) on Days 7 and 14, respectively, around the transplanted tumor. Then, the mice were

**Figure 7 (continued)**
euthanized at Day 28, and the xenograft tumors were harvested. (B) The growth curve of xenograft tumor size in nude mice was measured every three days ($n = 5$, one-way ANOVA followed by Tukey's multiple comparisons; $*$ vs Control, $***P < 0.001$; # vs CCL25, $###P < 0.001$). (C) Comparison of tumor tissue weights in the four groups ($n = 5$, one-way ANOVA followed by Tukey's multiple comparisons; $*$ vs Control, $***P < 0.001$; # vs CCL25, $###P < 0.001$). (D) Representative images from the histological staining of HE, TUNEL and Ki67 in tumor tissues. (E) Quantitative analysis of TUNEL and Ki67 staining (n = 5, one-way ANOVA followed by Tukey's multiple comparisons; $*$ vs Control, $*P < 0.05$, $**P < 0.01$; # vs CCL25, $##P < 0.01$, $###P < 0.001$). (F) Proliferation- and apoptosis-related transcription factors were detected in tumor tissues by WB. (G) Quantitative analysis of protein expression in (F) ($n = 5$, one-way ANOVA followed by Tukey's multiple comparisons; $*$ vs Control, $***P < 0.001$; # vs CCL25, $###P < 0.001$). (H) mRNA expression of proliferation- and apoptosis-related factors was measured by RT–qPCR ($n = 3$, one-way ANOVA followed by Tukey's multiple comparisons; $*$ vs Control, $**P < 0.01$, $***P < 0.001$; # vs CCL25, $###P < 0.001$). Data are presented as the mean ± SD.

its unique ligand CCL25, resulting in the activation of G protein and changes in cell biological behavior (*Atanes et al., 2020*; *Wu et al., 2021*). In recent years, CCR9 has been found in many cancers and proven to promote the malignant development of tumors once it interacts with its ligand CCL25 (*Tu et al., 2016*; *Xu et al., 2020*). However, the function of CCR9 and its related signaling pathway in SACC is unclear. This study first found that the expression of CCR9 in SACC was significantly upregulated compared with that in adjacent normal salivary glands. In histomorphology, there are three pathological types of SACC, including cribriform, tubular, and solid, in which the differentiation of malignancy ranges from low to high (*Morita et al., 2021*). SACC can be classified into three histological grades according to the percentage of solid tumor components. Grade I refers to tumors with tubular and cribriform areas but no solid components, Grade II refers to cribriform tumors with less than 30% of mixed solid areas, and Grade III refers to predominantly solid tumor. Histological grade is one of the important prognostic factors of SACC (*Perzin, Gullane & Clairmont, 1978*; *Szanto et al., 1984*). We found that the expression of CCR9 in cribriform and tubular SACC was significantly lower than that in solid SACC, which suggests that there is a certain correlation between CCL25/CCR9 and the malignant progression of SACC.

The CCR9/CCL25 axis could promote the proliferation of lung cancer cells by inhibiting the apoptosis of tumor cells (*Li et al., 2015*). Crystal, *Johnson-Holiday et al. (2011)* reported that the CCR9/CCL25 axis could induce different types of matrix metalloproteinases in breast cancer cells, and various MMPs promote the degradation of the extracellular matrix, thus creating favorable conditions for the invasion and metastasis of breast cancer cells. In this study, the results indicated that Ki67 and vimentin were strongly positive in SACC with high expression of CCR9, while E-cadherin was negative. Ki67 is mainly used to label cells in proliferation cycles, and its function is closely related to mitosis (*Smith et al., 2020*). Vimentin and E-cadherin are markers of EMT in tumor cells. Downregulation or deletion of E-cadherin expression leads to an increase in cell activities, while vimentin is an intermediate filament protein existing in mesenchymal cells, and its increased expression represents the occurrence of EMT in tumor cells (*Brabletz et al., 2018*; *Tian et al., 2020*). *In vitro*, we administered CCL25 to SACC-LM cells and observed changes in their biological behavior. We found that CCL25 significantly enhanced the proliferation and migration of SACC-LM cells, and the malignant behavior of tumor cells could be inhibited by CCR9 inhibitor (Vercirnon). Vercirnon is a selective antagonist of CCR9 and binds to

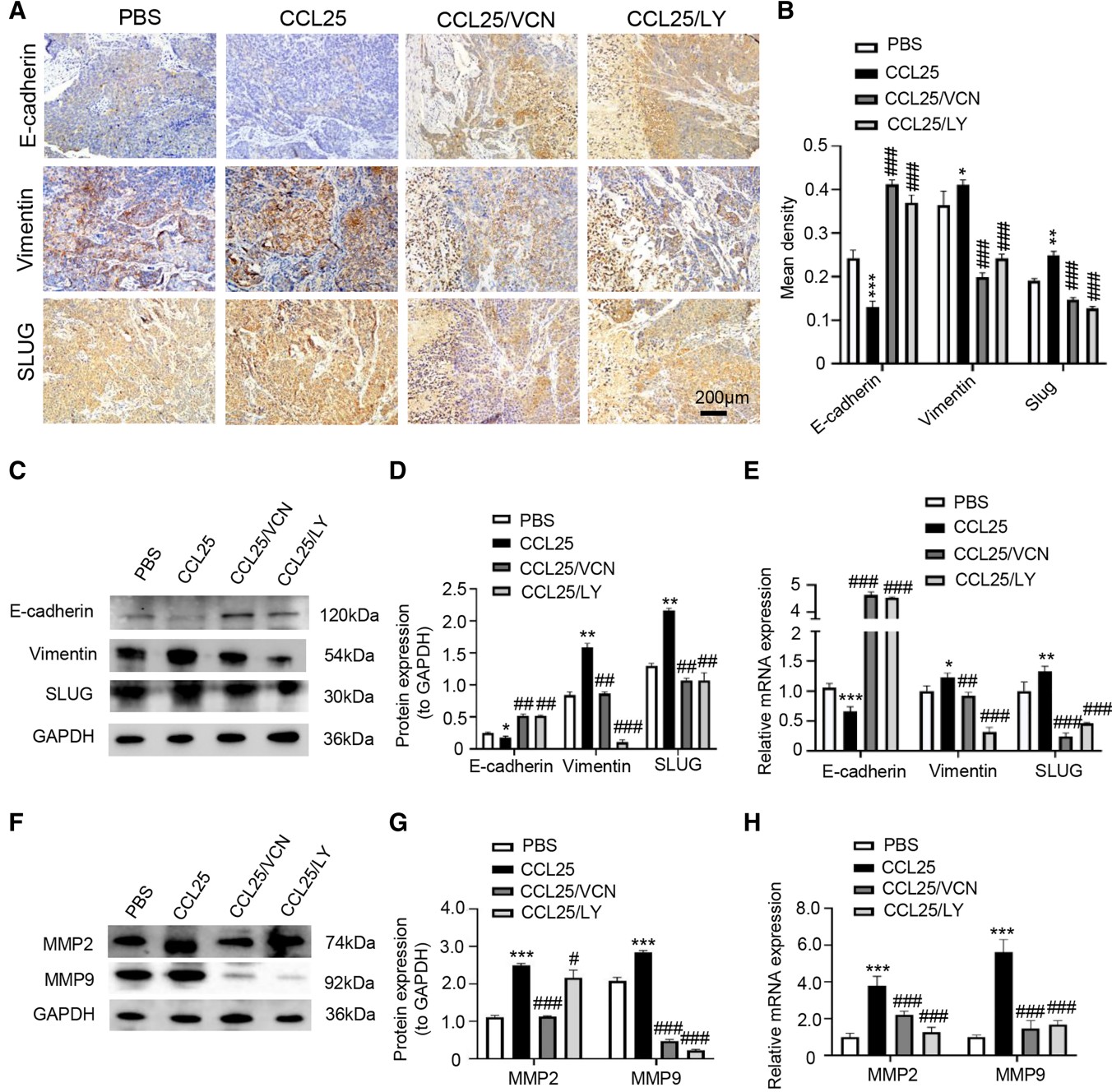

**Figure 8 CCL25/CCR9 promotes invasion and migration in a SACC xenograft mouse model by activating the PI3K/AKT signaling pathway.**
(A) The expression of tumor invasion- and metastasis-related factors (E-cadherin, vimentin and SLUG) in xenograft tumor tissues was detected by IHC staining. (B) The expression level was analyzed quantitatively ($n = 5$, one-way ANOVA followed by Tukey's multiple comparisons; * *vs* Control, *$P < 0.05$, **$P < 0.01$, ***$P < 0.001$; # *vs* CCL25, ###$P < 0.001$). (C, D) The protein expression of E-cadherin, vimentin and SLUG in xenograft tumors was measured by WB and statistically analyzed ($n = 5$, one-way ANOVA followed by Tukey's multiple comparisons; * *vs* Control, *$P < 0.05$, **$P < 0.01$; # *vs* CCL25, ##$P < 0.01$, ###$P < 0.001$). (E) The mRNA expression of E-cadherin, vimentin and SLUG in xenograft tumors was assessed by RT–qPCR ($n = 3$, one-way ANOVA followed by Tukey's multiple comparisons; * *vs* Control, *$P < 0.05$, **$P < 0.01$, ***$P < 0.001$; # *vs* CCL25, ##$P < 0.01$, ###$P < 0.001$). (F, G) The protein expression of ECM-related proteins (MMP2 and MMP9) in xenograft tumors and statistical analysis ($n = 5$, one-way ANOVA followed by Tukey's multiple comparisons; * *vs* Control, *$P < 0.05$, ***$P < 0.001$; # *vs* CCL25, ###$P < 0.001$). (H) The mRNA expression levels of MMP2 and MMP9 were measured by RT–qPCR ($n = 3$, one-way ANOVA followed by Tukey's multiple comparisons; * *vs* Control, ***$P < 0.001$; # *vs* CCL25, ###$P < 0.001$). Data are presented as the mean ± SD.

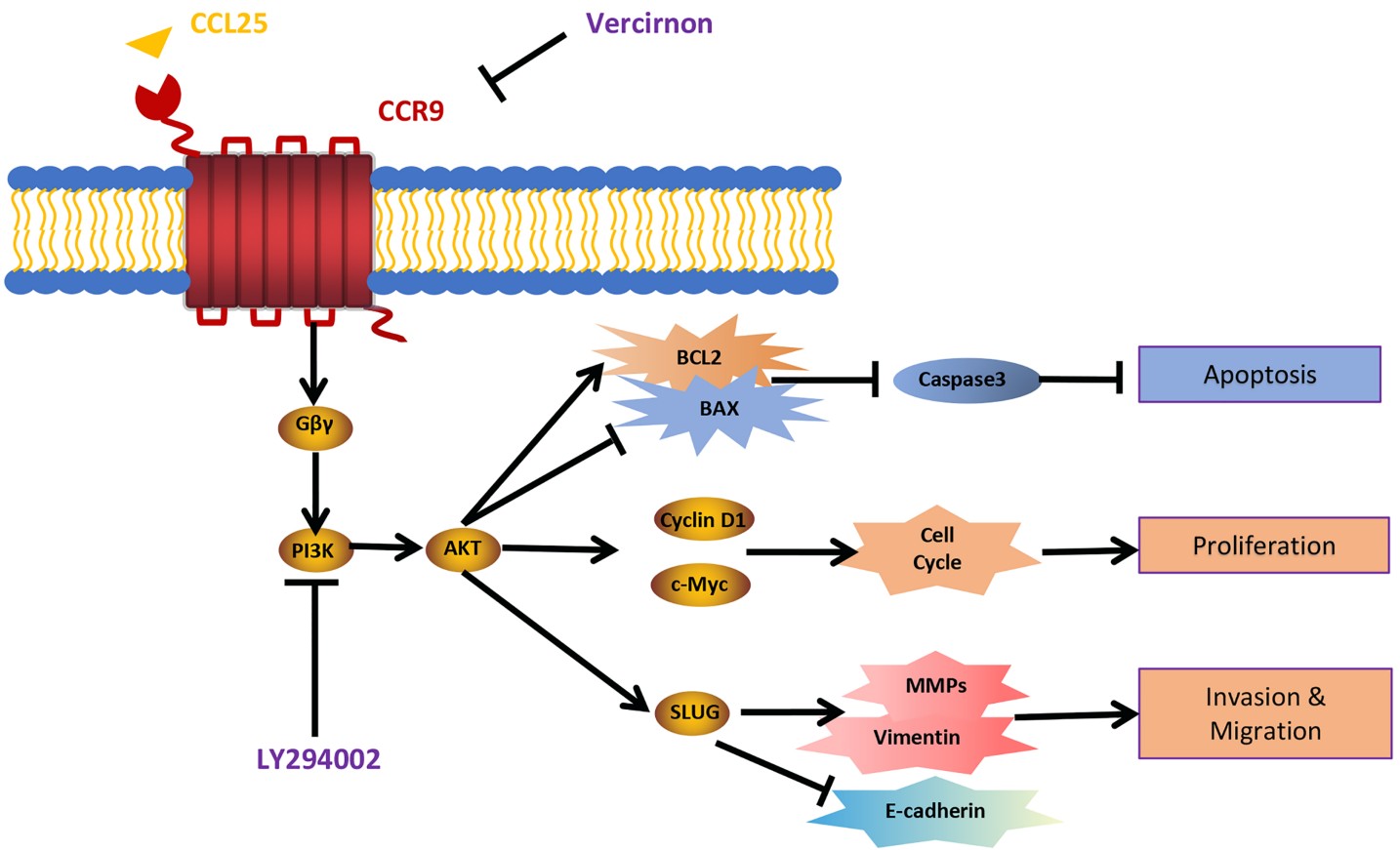

**Figure 9 Schematic model illustrating the PI3K/AKT pathway associated with CCRL25/CCR9-induced malignant biological behavior in SACC.** CCL25 acts on CCR9, a G protein-coupled receptor located on the SACC cell membrane, resulting in activation of the G protein and release of the $G_{\beta\gamma}$ subunit, thereby activating the PI3K/AKT signaling pathway. Vercirnon is a CCR9 inhibitor, and LY294002 is a blocker of the PI3K/AKT pathway. PI3K/AKT pathway activation can upregulate the mitochondrial-mediated anti-apoptotic factor BCL2, thus inhibiting caspase 3-dependent tumor cell apoptosis. Furthermore, CCL25 activation of the PI3K/AKT pathway leads to an increase in the expression of the cell cycle regulatory protein cyclin D1 and the protooncogene c-Myc, thus promoting the proliferation of tumor cells. Furthermore, the activation of the PI3K/AKT pathway activates its downstream transcription factor SLUG, leading to regulation of the expression of EMT-related factors (E-cadherin and vimentin) and ECM degradation-related factors (MMPs), which enhances the migration and invasion abilities of SACC cells.

the intracellular side of the receptor, exerting allosteric antagonism and preventing G-protein coupling (*Oswald et al., 2016*). Vercirnon could block the downstream signal transduction by inhibiting the interaction between CCL25 and CCR9, and even down regulate the expression level of CCL25. Therefore, our results indicated that the CCL25 could enhance the proliferation, invasion and metastasis of SACC cells through interaction with CCR9.

The malignant biological behavior of tumors mediated by the interaction of CCL25 and CCR9 is complex. CCR9/CCL25 can activate PI3K to phosphorylate AKT protein. The activation of the PI3K/AKT signaling pathway can regulate its downstream transcription factors NF-κB, β-catenin, cyclin D1 and c-Myc to promote the proliferation of tumor cells (*Wang et al., 2019*). In addition, the JAK/STAT pathway activated by CCL25 can alleviate the antitumor ability of T cells and promote the immune escape of tumor cells

(*Soldevila et al., 2004*; *Tu et al., 2016*). CCR9/CCL25 can also activate Ras/Raf/MARK and RhoA/Rock signals to strengthen the degradation of ECM and change the cytoskeleton arrangement, thus enhancing tumor cell motility (*Golec, Henao Caviedes & Baldwin, 2016*). Our results showed that the interaction of CCL25 and CCR9 could significantly upregulate the S473 phosphorylation level of AKT in SACC cells, indicating that AKT was activated. When PI3K/AKT was inhibited under CCR9/CCL25 activation, the proliferation of SACC cells was obviously attenuated. Moreover, PI3K/AKT inhibitor significantly decreased the expression of the proliferation-related factors Ki67, cyclin D1 and c-Myc and the level of the antiapoptotic factor BCL2 in SACC cells treated with CCL25, but the level of these related factors could not be further reduced by CCR9 inhibitor. In xenograft mouse models, we further proved that CCL25 could effectively promote tumor growth and suppress tumor apoptosis, and this enhancement could be reduced by inhibitors of CCR9 and PI3K/AKT, indicating that the PI3K/AKT signaling pathway played a key role in CCL25/CCR9 process.

Activation of the PI3K/AKT signaling pathway can not only promote proliferation and anti-apoptosis but can also enhance the migration, invasion and drug resistance of tumor cells (*Han et al., 2018*). There are many downstream targets of PI3K/AKT induced by CCL25. *Deng et al. (2017)* found that in adult T-cell acute lymphoblastic leukemia, CCL25/CCR9 could activate PI3K/AKT/RhoA signaling by upregulating Wnt5a, enhancing the polymerization of actin and the formation of cell pseudopodia, thus playing a key role in promoting tumor invasion and migration. *Wei et al. (2020)* indicated that the activation of PI3K/AKT in colon cancer cells could inhibit GSK3β, which reduced the degradation of SLUG and enhanced the EMT of tumor cells. Our results found that in SACC, the activation of the PI3K/AKT signaling pathway mediated by CCL25 could obviously promote the EMT process. SLUG, a zinc-finger transcription factor, can bind to E-box elements in the E-cadherin-specific promoter region, repress the expression of E-cadherin and enhance the process of EMT in tumor cells (*Bai et al., 2017*). Other studies suggested that SLUG was not involved in the downregulation of E-cadherin but promoted EMT by maintaining the phenotype of mesenchymal cells (*Sterneck, Poria & Balamurugan, 2020*). In this study, we found that CCL25 could significantly upregulate the expression of SLUGand SNAIL1 in SACC cells, especially SLUG. When PI3K/AKT inhibitors were added to SACC cells administered CCL25, the expression level of SLUG was obviously reduced, indicating that the interaction between CCL25 and CCR9 could regulate the level of SLUG by activating the PI3K/AKT signaling pathway. The specific role of SLUG mediated by CCL25/CCR9 in SACC cells need to be further explored.

Tumor invasion and metastasis are complex processes, and one of the key steps is the degradation of the extracellular matrix. Many studies have proven that matrix metalloproteinases (MMPs) can degrade the extracellular matrix and promote tumor invasion and metastasis (*Cox, 2021*; *Stallings-Mann et al., 2012*). It has been reported that in highly invasive tumors, the expression of MMP2 is increased in tumor cells with high expression of SLUG, suggesting that the relationship between SLUG and MMPs might play a potential role in tumor invasion and the EMT process (*Horejs et al., 2017*; *Li et al., 2014*). In this study, the results showed that CCL25/CCR9 could increase the expression of

MMP2 and MMP9 by activating PI3K/AKT in SACC, which was confirmed in xenograft tumor mice, suggesting that CCL25/CCR9-activated PI3K/AKT might modulate MMPs through the transcription factor SLUG. However, the specific molecular mechanism by which SLUG regulates MMPs in SACC needs to be further investigated.

## CONCLUSION

This study proved that CCR9 was highly expressed in SACC, and its expression was significantly correlated with the malignant biological behavior of tumors. In *in vitro* experiments, we found that the interaction between CCL25 and CCR9 could activate the PI3K/AKT signaling pathway, regulate its downstream factors and promote the proliferation, migration and invasion of SACC cells. *In vivo*, we used inhibitors of CCR9 and PI3K/AKT to further confirm the effectiveness and feasibility of SACC therapy by targeting the interaction of CCL25 and CCR9. This study provides a new potential target and strategy for the clinical treatment of SACC.

## ACKNOWLEDGEMENTS

The authors would like to thank Professor Tingjiao Liu for giving SACC cell lines as a gift.

### Funding

This work was supported by grants from the Natural Science Foundation of China (No. 81802706 to Lu Gao; No. 81771032 to Fu Wang), the Scientific Foundation of Education Department of Liaoning Province (No. LZ2020035 to Lu Gao), and the Natural Science Foundation of Liaoning Province (No. 2021-MS-293 to Lu Gao; No. 20180550028 to Songling Chai). The funders had no role in study design, data collection and analysis, decision to publish, or preparation of the manuscript.

### Grant Disclosures

The following grant information was disclosed by the authors:
Natural Science Foundation of China: 81802706, 81771032.
Scientific Foundation of Education Department of Liaoning Province: LZ2020035.
Natural Science Foundation of Liaoning Province: 2021-MS-293, 20180550028.

### Competing Interests

The authors declare that they have no competing interests.

### Author Contributions

- Songling Chai performed the experiments, analyzed the data, prepared figures and/or tables, and approved the final draft.
- Zhihao Wen performed the experiments, analyzed the data, prepared figures and/or tables, and approved the final draft.
- Rongxin Zhang performed the experiments, analyzed the data, prepared figures and/or tables, and approved the final draft.

- Yuwen Bai performed the experiments, analyzed the data, prepared figures and/or tables, and approved the final draft.
- Jing Liu performed the experiments, analyzed the data, prepared figures and/or tables, and approved the final draft.
- Juanjuan Li performed the experiments, analyzed the data, prepared figures and/or tables, and approved the final draft.
- Wenyao Kongling performed the experiments, analyzed the data, prepared figures and/or tables, and approved the final draft.
- Weixian Chen performed the experiments, analyzed the data, prepared figures and/or tables, and approved the final draft.
- Fu Wang conceived and designed the experiments, performed the experiments, analyzed the data, prepared figures and/or tables, authored or reviewed drafts of the article, and approved the final draft.
- Lu Gao conceived and designed the experiments, performed the experiments, analyzed the data, prepared figures and/or tables, authored or reviewed drafts of the article, and approved the final draft.

### Human Ethics

The following information was supplied relating to ethical approvals (*i.e.*, approving body and any reference numbers):

This study was approved by the Ethics Association of Dalian Medical University.

### Animal Ethics

The following information was supplied relating to ethical approvals (*i.e.*, approving body and any reference numbers):

Dalian Medical University provided full approval for this research (aee20017).

### Data Availability

The raw data is available in the Supplemental Files.

### Supplemental Information

Supplemental information for this article can be found online at http://dx.doi.org/10.7717/peerj.13844#supplemental-information.

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
