# Peer review of "CCL25/CCR9 interaction promotes the malignant behavior of salivary adenoid cystic carcinoma via the PI3K/AKT signaling pathway"

_PeerJ, doi:10.7717/peerj.13844_

## Round 0.1 · original submission · Major Revisions

After careful consideration, we feel that it has merit, but is not suitable for publication as it currently stands. Therefore, my decision is "Major Revision."

We invite you to submit a carefully revised version of the manuscript that addresses all the points raised by the three reviewers. If the paper can be substantially revised to take account of these comments, I would be happy to consider it for the next round of review. I must stress that each comment should be considered and it would be very useful if you could detail, in a covering letter, how you have answered each point.

Reviewer 1 ·

Basic reporting

1. Gene symbols in this paper should be standardized. For example, “Slug” should be only used in mouse and “SLUG” in human.
2.The language should be improved.
3.There were some typos (eg. In Fig. 1 E, the R2 value of E-Cadherin should be negative based on the results.).

Experimental design

Songling Chai et al. elaborated on the effects and mechanisms of CCL25/CCR9 interaction in salivary adenoid cystic carcinoma (SACC). They found that CCR9 expression levels were increased in SACC which was associated with tumor proliferation and metastases. They further found that CCL25/CCR9 promoted SACC development via activating the PI3K/Akt pathway in vitro and in vivo.

This study provides some insights on the functions and mechanism of CCL25/CCR9 in SACC.
But there are still some comments as below:

1 The authors should increase the SACC specimen numbers in Fig.1 A to get more sufficient evidence.

2. The authors should also check the protein expression levels by WB in Fig.1B.

3.The authors should provide the same IHC image regions to compare the expression levels of different targets in Fig.1 D.

4. The authors should employ other assays such as EdU Staining to double check the CCL25/CCR9 effects on proliferation. Also, the authors should also check the protein expression levels of downstream targets in Fig.2.

5. The authors should provide more data to support the CCL25/CCR9 functions of activating PI3K/Akt signaling pathway such as loss of function assay.

6. In Fig.5, the authors should provide direct evidence to support the effects of CCL25/CCR9 interaction on apoptosis such as TUNEL assay.

7. In Fig.7, the authors should provide the original images of xenograft tumors. Also, the authors should show the same image regions when comparing HE, TUNEL and Ki67 staining results as well as Fig. 8A.

8. The figure resolutions should be improved throughout the whole manuscripts especially for the image of WB and staining.

Validity of the findings

no comment

Reviewer 2 ·

Basic reporting

The background information introduced the current knowledge of CCR9-CCL25 in many tumors but its role in salivary adenoid cystic carcinoma is unknown. This part was clearly presented by the author.
The language is easy to follow. The structure and data format are according to the standards of PeerJ.

Experimental design

The experiments design was relevant, meaningful and within the scope of the journal.
Technical and ethical statements met the journal and were described with sufficient detail in the Method section.
Here are some comments for the author to further strengthen the findings.
a. Most of the downstream targets were only tested on mRNA levels (Figure 2C-D, Figure 3E, Figure 5C-H). Some genes didn’t have large fold changes on qPCR(Figure 2 cMyc in CCL25NCN vs. CCL25). It would be good to verify them on protein levels by western blot. In addition, the human gene symbols should be capitalized and italicized to distinguish them from protein names. For example, Slug should be SNAIL2.
b. Figure 4, the phosphorylation sites for AKT and STAT3 should be pointed out here. Since there are more than one pAKT or pSTAT3 antibodies which should be clarified the sites that the author tested in the experiments. In addition, to evaluate the changes of phosphorylated proteins, using GAPDH is not enough as a control. Total AKT, ERK and STAT3 are the more common controls in similar assessments.
c. In Figure 4A, 4B, Figure 5-7, since the PI3K-AKT pathway is so important for the survival of any type of cells. I would suggest here adding a combined group of CCL25 inhibitor and PI3K inhibitor to see whether you could observe an adds-on effect. If the combined effects were much stronger than any individual inhibitor, then CCR9-CCL25 and PI3K/AKT may only share partial pathway in SACC malignancy. If the combined effects were almost the same as any individual inhibitor, then it indicates that CLL25 mainly dependent on PI3K/AKT pathway to confer malignancy.
d. One novel finding from this study is that the author found Slug is induced by CCL25 but reduced by PI3K/AKT pathway inhibitor. The author could further expand and verify this finding by overexpressing Slug or knockout Slug in SACC cell line and repeating the migration and invasion experiments to verify the importance of Slug in CCL25 induced malignancy regulation.
e. Could you please explain why you only choose MMP2/9 from MMP family to assess SACC metastasis in your experiment? Certain reference may be cited to justify the importance of these two in SACC in terms of invasion and metastasis.

Validity of the findings

The author addressed the mayor scientific question with well designed experiments and
robust, statistically sound &controlled data. Conclusions are also well stated.
Some minor comments:
a. One minor thing, are the findings only applied to CCR9 high SACC? Most of the experiments were only assessed in SACC-LM cells. Have you tried the SACC-83?
b. Figure 7D, the blue color for Tunnel staining is too difficult to be seen.
c. Did you see tumor metastasis to other organs in your in vivo model?
d. It would be better if statistic figures could be shown with overlay bar and scatter plots.
e. The western blots in the raw data file do not have markers.
f. “in vivo”, “in vitro” should be italicized.

Additional comments

Although most of the link, such as the relationship between CCR9-CCL25 and tumor malignancy, CCR9-CCL25 and PI3K/ATK pathway are not the first report. However, the author clearly stated the relationship with sufficient data of their relationships in SACC, which has not been fully reported. In addition, the manuscript is generally well written and easy to follow.

·

Basic reporting

1. In the first figure, the article has shown 3 different types of SACC. In my opinion, you should talk a little about the 3 types in your article.
2. If you can show the type of SACC in figures 1 D and E, that would be perfect.
3. In figure 1 C, the author chooses three different cell lines to compare the CCR9, but figures 2 A and B only have two cell types, and after these only have one cell type of SACC-LM. I hope to know what strategy was the author based on.
4. In my opinion, the article has a little inadequate in discussing the interaction of CCL25/CCR9. Such as, the CCL25 gene expression would upgrade or down when treated by CCR9’s inhibitor.

Experimental design

Normally, we treated by A and B would have 4 groups (control, A, B, A plus B.). Why are your two treatments of CCL25/LY and CCL25/VCN only have 3 groups, not 4 groups?

Validity of the findings

The results partially have been reported, such as ST type was high CCL25.

---

## Round 0.2 · accepted · Accept

Dr. Gao, Congratulations! Per the reviewers' decisions, we are pleased to inform you of the acceptance of your manuscript. Thanks for choosing PeerJ to publish your scientific work.

I look forward to cooperating with you in the future.

Reviewer 1 ·

Basic reporting

The results of the manuscript strongly supported their conclusion. The language is good to follow.

Experimental design

The experimental design is sufficient to be published in PeerJ after revision.

Validity of the findings

no comment

Reviewer 2 ·

Basic reporting

The authors have greatly improved the manuscript.

Experimental design

The authors have perform additional experiments which could answer the questions raised by the reviewer.

Validity of the findings

No comment

Additional comments

The human gene symbols should be capitalized and italicized to distinguish them from protein names. The authors didn't correct the label of genes in the figures.

·

Basic reporting

Congratulation, the article can be accepted.

Experimental design

no comment

Validity of the findings

no comment